# Gut microbiome and metabolome profiles in renal allograft rejection from multiomics integration

Xing Dai,[1] Yu Cao,[2] Lin Li,[3,4] Yue-Xin Gao,[1] Jian-Xi Wang,[3,4] Yao-Juan Liu,[3,4] Ting-Ting Ma,[5] Jian-Ming Zheng,[2] Pan-Pan Zhan,[3,4] Zhong-Yang Shen[3,4]

**ABSTRACT**   The gut microbiome and metabolome play crucial roles in renal allograft rejection progression. Integrated multiomics analyses may provide a comprehensive understanding of specific underlying mechanisms, which remain elusive. This study aimed to identify new approaches for clinical renal allograft rejection diagnosis and treatment. Thirty-five patients were divided into three groups: the rejection ($n$ = 16), dysfunction ($n$ = 7), and control ($n$ = 12) groups. Metagenomic sequencing and nontargeted metabolomics were used to analyze stool and plasma samples. Significant microbiota, metabolites, and signaling pathways were identified. LASSO regression was used to construct a diagnostic model, and its diagnostic value was assessed via receiver operating characteristic curves. The microbiota composition and the related genes in the rejection group significantly differed from that in the dysfunction and control groups at the phylum, genus, and species levels ($P < 0.001$). The core species in the rejection group networks were *Escherichia coli* and *Ruminococcus gnavus*, while core species in the dysfunction group networks were *Faecalibacterium prausnitzii* and *Bacteroides ovatus*. The balance of specific microbial species was associated with kidney function in rejection patients. Spearman analysis revealed that specific differential species like *Agathobaculum butyriciproducens* and *Gemmiger qucibialis* were closely linked to the levels of serum 4-pyridoxic acid, 4-acetamidobutanoate, and fecal tryptamine from specific differential pathways. Finally, we constructed four clinical models to distinguish the rejection and dysfunction groups, and the model had excellent diagnostic performance. Altered gut microbiota may contribute to changes in metabolic pathway activity and metabolite abundance in rejection and dysfunction patients, which are strongly correlated with host immunological rejection. The diagnostic model, developed based on the gut microbiota and metabolites, has high clinical value for diagnosing renal rejection.

**IMPORTANCE** This study aimed to screen new markers for non-invasive diagnosis by the gut microbiome and metabolome analysis, providing new insights into rejection mechanisms and identifying new approaches for clinical renal allograft rejection diagnosis.

**KEYWORDS** kidney transplantation, rejection, gut microbiome, metabolome, prediction model

**Peer Reviewers** Ali Chaari, Weill Cornell Medicine, Doha, Qatar; Somya Aggarwal, Jawaharlal Nehru University, New Delhi, India; Aida G. Gabdoulkhakova, Kazanskij federal'nyj universitet Institut fundamental'noj mediciny i biologii, Kazan, Russia

Address correspondence to Jian-Ming Zheng, zhengjm317@126.com, Pan-Pan Zhan, zpopo72@163.com, or Zhong-Yang Shen , zhongyangshen@vip.sina.com.

The authors declare no conflict of interest.

See the funding table on p. 15.

Allograft rejection is a serious and common postoperative complication that affects the survival rate of both the graft and the patient (1). Currently, puncture biopsy is the gold standard for diagnosing postoperative allograft rejection, but it presents a high risk of complications, and patient compliance is poor (2). Thus, a noninvasive method for predicting and diagnosing rejection after kidney transplantation is urgently needed. Now, there are several types of non-invasive biomarker technologies (metagenomics, metabolomics, epigenetics, proteomics, and transcriptomics) currently used, but it is

not clear which method is more accurate. Recent study revealed that gut microbiota is associated with organ post-transplant complications, like diarrhea, urinary tract infection, rejection, and mortality (3–7). Obviously, gut microbiota plays an important role after solid organ transplantation. It may become an insightful biomarker to predict graft rejection because gut microbiota can modulate the immune system through metabolites that affect graft function (8, 9). The studies have suggested an association between allograft rejection and microbiota dysbiosis, noting that the gut microbiota is disturbed after transplantation; this disturbance in microbial metabolites promotes rejection (10). After transplantation, the integrity of the intestinal epithelium is compromised, allowing bacteria and their metabolites to enter the internal environment (11). This dysregulation induces a proinflammatory response in intestinal epithelial cells. The IL-1, IL-6, and IL-18 secreted by intestinal epithelial cells are triggered to eliminate pathogens. Dendritic cells and macrophages induce the development of the effector CD4$^+$ T cells TH1 and TH17 by secreting IL-10 and IL-23 (12). Simultaneously, gut-derived toxic metabolites—phenols and indoles are further metabolized into P-cresol sulfate (PCS), which enters the systemic circulation. PCS accumulation in renal tubule cells triggers reactive oxygen species production, resulting in the production of inflammatory cytokines and profibrotic factors and subsequent cell damage (12). Because of the strong correlation between the gut microbiota and metabolites, the combination of microbiota and metabolomics may improve the accuracy of diagnosis of allograft rejection. Therefore, to describe the gut microbiota and metabolite composition in renal rejection patients and their link to disease progression, serum and fecal samples were used to generate microbial and metabolic profiles. The aim of this study was to explore the mechanisms by which the gut microbiota and metabolites impact renal allografts and suggest new noninvasive diagnostic approaches.

## MATERIALS AND METHODS

### General information

The inclusion criteria for patients were as follows: (i) aged 18–65 years, (ii) first orthotopic kidney transplantation, (iii) negative lymphocytotoxicity test results, (iv) uniform perioperative antibiotic use, and (v) uniform immunosuppressive therapy.

The exclusion criteria were as follows: (i) intestinal inflammatory diseases, (ii) recent diarrhea, and (iii) unique dietary habits.

According to these criteria, 35 patients admitted to Tianjin First Central Hospital affiliated with Tianjin Medical University from May 2023 to October 2024 were included.

### Grouping

Patients were divided into three groups: the rejection group ($n$ = 16, patients with postoperative renal puncture biopsy results showing rejection), the dysfunction group ($n$ = 7, patients with postoperative renal puncture biopsy results showing no rejection), and the control group ($n$ = 12, patients with normal postoperative renal function) (Fig. 1). Rejection was defined according to the 2019 Banff criteria (13).

### Observed indicators

Baseline data were collected from the electronic medical records system of Tianjin First Central Hospital (Table S1), including (i) donor type, sex, age, warm ischemia time (WIT), and cold ischemia time (CIT); (ii) recipient sex, age, body mass index (BMI), time after kidney transplantation (months), urea level, creatinine (CREA) level, urea/CREA ratio, uric acid (UA), urinary protein/creatinine ratio (UPRO/CREA), estimated glomerular filtration rate (eGFR), panel-reactive antibody (PRA) levels, human leukocyte antigen (HLA) mismatch number, intraoperative antibiotic use, intraoperative induction regimen, postoperative immunosuppressive regimen; and (iii) recipient infection information.

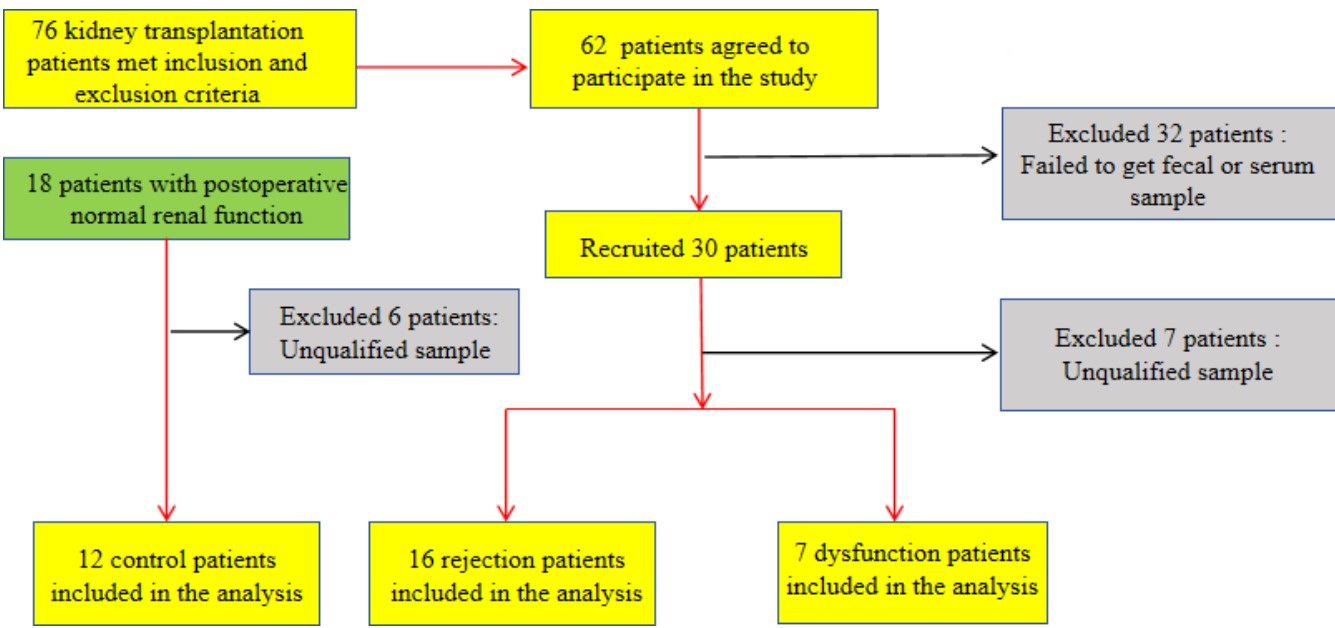

**FIG 1** Overview of patient group allocation.

## Sample information

Fecal samples were collected the day before the patient's biopsy, frozen at −20°C, and later transferred to −80°C for long-term storage. Each sample was divided into two parts: one for metagenomic sequencing and one for gut metabolomics analysis via nontargeted liquid chromatography−mass spectrometry (LC-MS). Serum samples were collected the day before biopsy and centrifuged at 3,000 RPM and 4°C; the supernatant was then removed and frozen at −20°C before being transferred to −80°C for long-term storage. Metabolomic analysis was conducted via nontargeted LC-MS. Microbiota and metabolite analyses were performed by BGI (Shenzhen, China).

## Statistical methods

All the statistical analyses were performed via SPSS Statistics for Windows, version 24.0. Normally distributed data are presented as the mean ± standard deviation, and independent-sample $t$ tests were used for intergroup comparisons. Skewed data are presented as medians (interquartile ranges), and the Mann–Whitney $U$ test was used for intergroup comparisons. Categorical data are presented as frequencies and percentages, and either the chi-square test or Fisher's exact test was used for comparison. Correlations among microbial, metabolomic were calculated using Spearman's correlation analysis. Variables associated with outcomes of rejection or dysfunction were screened using Lasso regression, with Lambda.min as the criterion. $P$ value < 0.05 (corrected by the Benjamini−Hochberg method) was considered to indicate statistical significance.

## Gut microbiota analysis

SOAPnuke software was used for filtering quality control of the original data, and Bowtie2 was used to compare and remove the host sequence to generate clean data. MEGAHIT was used for the assembly of sequences on the basis of K-MER and to produce contigs. Then, MetaGeneMark was used to predict gene sequences in the contigs. CD-HIT software was used to remove redundant genes, and Salmon software was used to determine the relative abundance of each gene. DIAMOND or RGI was used to compare nonredundant genes to those in the Kyoto Encyclopedia of Genes and Genomes (KEGG) database to complete gene function annotation. The sequence numbers of the species detected in the sample were identified by comparing the Kraken 2 results with the

screening NCBI NT database, and Bracken 2 was used to estimate the actual abundance of the species in the sample and complete the species annotation.

The Wilcoxon/Kruskal−Wallis test was used to compare abundance data for the microbial community. Shannon index was used for alpha diversity index, Kruskal-Wallis test was used for alpha diversity and beta diversity among groups, beta-diversity was calculated based on the Bray–Curtis distance, a box plot was drawn, and the distances among the three groups were determined. Linear discriminant analysis effect size (LEfSe) was used to identify the microbial taxa that significantly differed among rejection, dysfunction and control groups, with a threshold LDA score of >2 used. The differences among the three groups of KEGG Orthology (KO) genes were analyzed via stamp analysis. Gephi-0.9.2 was used to draw the intricate network.

## Gut metabolites

A Waters 2777C UPLC (Waters, USA) series Q Exactive HF high-resolution mass spectrometer (Thermo Fisher Scientific, USA) was used for the separation and detection of metabolites. The obtained data were normalized via the probabilistic quotient normalization method to obtain the relative peak area. Quality control (QC)-based robust LOESS signal correction was used to correct the batch effect. Compounds with a coefficient of variation (CV) of the relative peak area greater than 30% were removed from all the QC samples.

We screened differentially abundant metabolites according to the fold change (FC ≥ 1.2 or ≤ 0.83), $P$ value < 0.05, and variable importance in projection (VIP) value (≥1). The biochemical pathways of the differentially abundant metabolites were identified by searching the KEGG database, and the metabolites were classified according to their pathway involvement. Enrichment analysis was conducted for metabolites in a functional node.

## Multiomics integration

The network was drawn by Spearman correlation analysis to reveal the correlation between gut microbiota and metabolites. The gut microbiota and metabolites were linked by the annotated pathway in the network. It helps to explore the mechanisms by which the gut microbiota affects rejection reaction by metabolites, and the predictive effect of combined markers on rejection was analyzed.

## RESULTS

### Basic information

Significant differences were observed in the urea level, CREA level, urea/CREA ratio , and eGFR among the three groups (Table 1). Intraoperative antibiotic, intraoperative induction regimen, postoperative immunosuppressive regimen and recipient infection information were similar among the groups. Differences in recipient age, sex, BMI, sample collection times, number of HLA mismatches, PRA, ABO incompatibility, and donor age, sex mismatch, WIT, CIT and donor type were analyzed, and the results indicated no significant differences among the groups.

### Comparison of the gut microbiota structure and functional pathways among the three groups

At the phylum and genus levels, the alpha diversity of the gut microbiota was similar among the three groups ($P$ > 0.05; Fig. 2A), while the beta diversity, which was analyzed via the Bray-Curtis distance, differed among the three groups ($P$ < 0.05; Fig. 2A). At the family level, the rejection group exhibited increased *Firmicutes* and *Proteobacteria* abundances and decreased *Bacteroidota* abundance. The dysfunction group showed *Bacteroidota* expansion. At the genus level, increased *Escherichia* and *Faecalibacterium* abundances and decreased *Phocaeicola, Bacteroides,* and *Lachnospira* abundances were

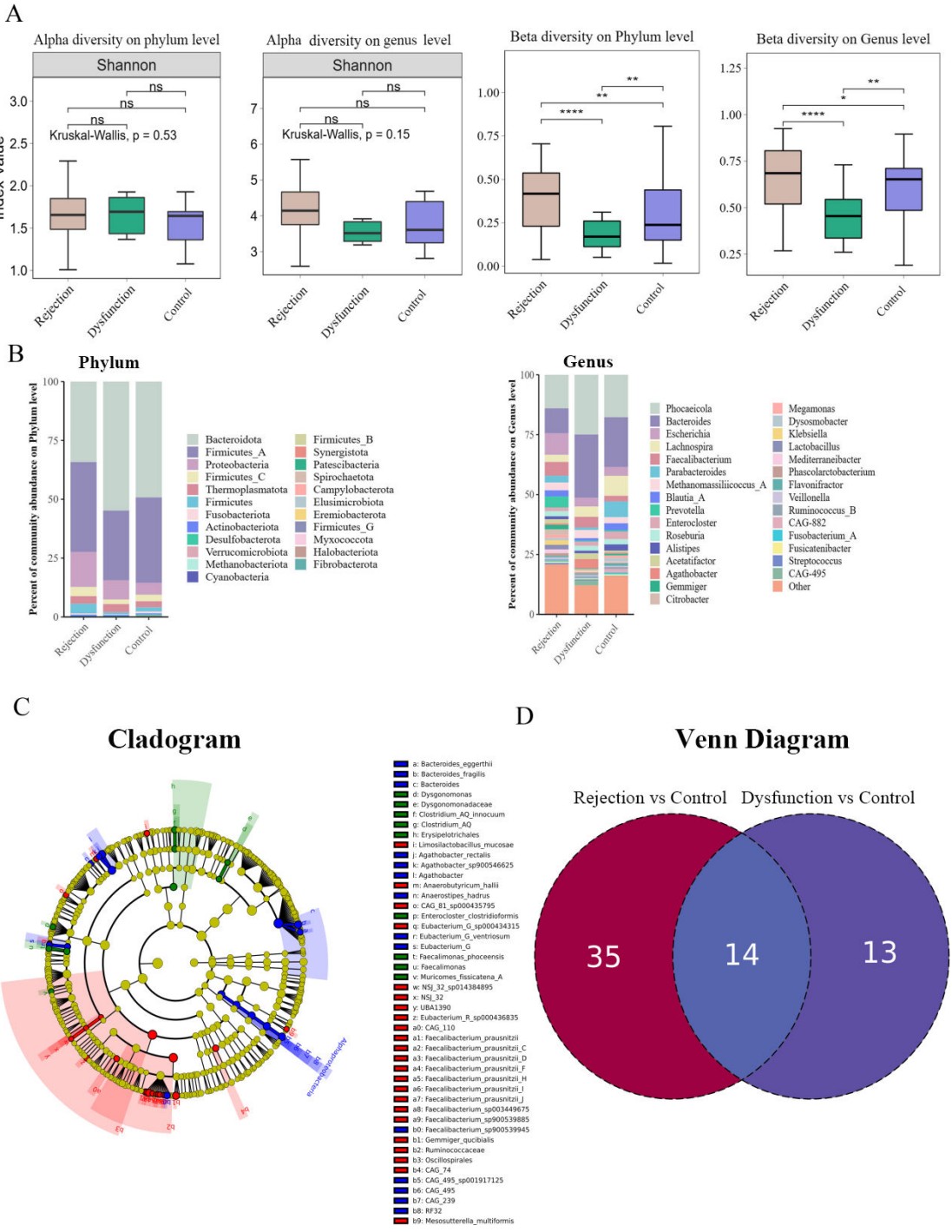

**FIG 2** (A) Alpha diversity showing differences on the phylum and genus level among the three groups, Beta diversity based on the Bray-Curtis distance showing differences on the phylum and genus level among the three groups. (B) Average relative proportions of the main phylum and genus in the three groups. (C) LEfSe shows essential differences in bacterial abundance (family to species level) among the three groups. Only LDA threshold values >2 are shown. (D) Venn diagram shows specific varied species in rejection and dysfunction groups.

observed in the rejection group. The dysfunction group presented high *Phocaeicola* and *Bacteroides* levels (Fig. 2B).

LEfSe analysis revealed significant increases in the abundances of *Oscillospirales, Ruminococcaceae, Gemmiger qucibialis,* and *Faecalibacterium prausnitzii* in the rejection group. In the dysfunction group, *Bacteroides* and *Agathobacter* were more enriched,

whereas *Erysipelotrichales, Dysgonomonadaceae,* and *Faecalimons* were more enriched in the control group (Fig. 2C).

At the species level, we observed the altered patterns that varied in the rejection or dysfunction group. It included 35 species in the rejection group (24 species increased and 11 species decreased) and 13 species in the dysfunction group (10 species increased and 3 species decreased) (Fig. 2D). The mean abundance of these species in the groups is shown (Fig. 3A and B). We identified specific altered KO genes in the rejection and dysfunction groups with respect to their functional pathways by the "Reporterscore" algorithm (14). The rejection group was characterized by three main KEGG pathways. The KO genes in the rejection group were enriched in Xenobiotics biodegradation and metabolism and decreased in Lipid metabolism and Metabolism of cofactors and vitamins, whereas the dysfunction group KO genes were enriched in Phosphonate and phosphinate metabolism and Starch and sucrose metabolism and were decreased in Amino acid metabolism (Fig. 3A and B).

## Bacterial co-occurrence network

The co-occurrence of gut microbes is a reflection of their interactions within an ecosystem. Spearman's correlation analysis was used to identify the 50 most abundant bacterial species in the three groups ($P < 0.05$ using the criterion of a correlation coefficient $>|0.5|$). This led to the identification of *Escherichia coli, Ruminococcus gnavus,* and *Escherichia sp00*0208585 in the rejection group; *Faecalibacterium prausnitzii, Bacteroides ovatus,* and *Parahacteroides distasonis* in the dysfunction group; and *Enterocloster bolteae* in the control group as core species (Fig. 3C). Interactions among microbes in the rejection group were rare, indicating that the rejection reaction may adversely affect some beneficial interactions of the gut microbiota. These data revealed different patterns in the bacterial co-occurrence networks among the three groups. *Escherichia coli* and *Ruminococcus gnavus* have been reported to be potential proinflammatory species (15, 16). However, *Bacteroides ovatus* is a short-chain fatty acid (SCFA)-producing species, whereas *Parahacteroides distasonis* has anti-inflammatory functions, indicating that the core species in the rejection group networks tended to be proinflammatory species, whereas those in the networks of the dysfunction group presented a combination of proinflammatory and SCFA-producing species (17, 18).

## The balance between healthy microbiota and dysbiosis is correlated with clinical indicators

The balance between a metabolically healthy microbiota and dysbiosis is important for maintaining host metabolic homeostasis. We evaluated the balance at the genus level between the rejection and control groups via the "selbal" algorithm (19). This finding revealed that *Faecalomonas* was the most prevalent species in the balance of the cross-validation between the two groups (Fig. 4A). Patients with rejection had higher balance scores than the controls, indicating the presence of higher relative abundances of *Klebsiella, Citrobacter,* and *Anaerostipes* in the rejection group than in the control group; the discrimination value of the determined balance was also important, with an apparent AUC value of 0.948. Furthermore, the associations of the gut microbial balance with clinical biochemical parameters were evaluated, which revealed that *Clostridium* and *Enterocloster* were the bacteria most closely related to kidney function (Fig. 4B). Higher balance scores were associated with higher urea, CREA and eGFR. The ratio of lower abundance of *Dysgonomonas* to higher *Oscillospiraceae* ER4 was associated with both urea ($R^2 = 0.539$; Fig. 4C) and CREA ($R^2 = 0.435$;, Fig. 4C), and the ratio of lower abundance of *Dorea* to higher *Dysgonomonas* was associated with GFR ($R^2 = 0.398$; Fig. 4C). Thus, the balance of specific microbial genera was associated with kidney function in patients with rejection.

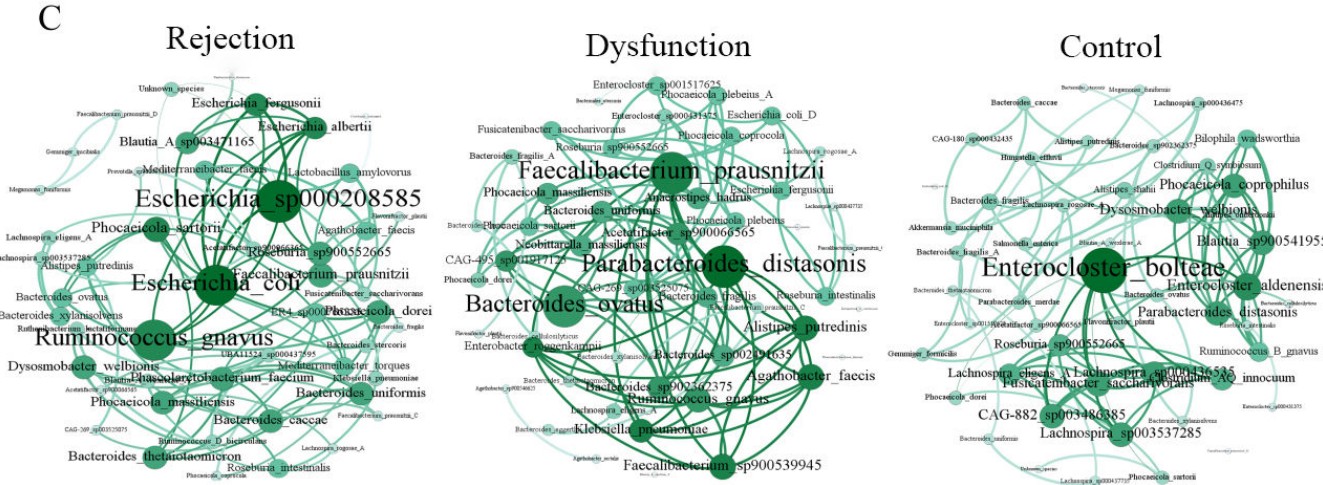

**FIG 3** (A) Gut microbial species with differentiated alterations between the rejection and control groups; the right part is the pathway with differentiated KO genes annotated. (B) Gut microbial species with differentiated alterations between the dysfunction and control groups; the right part is the pathway with differentiated KO genes annotated. (C) Co-occurrence network of gut microbial genus in patients among three groups.

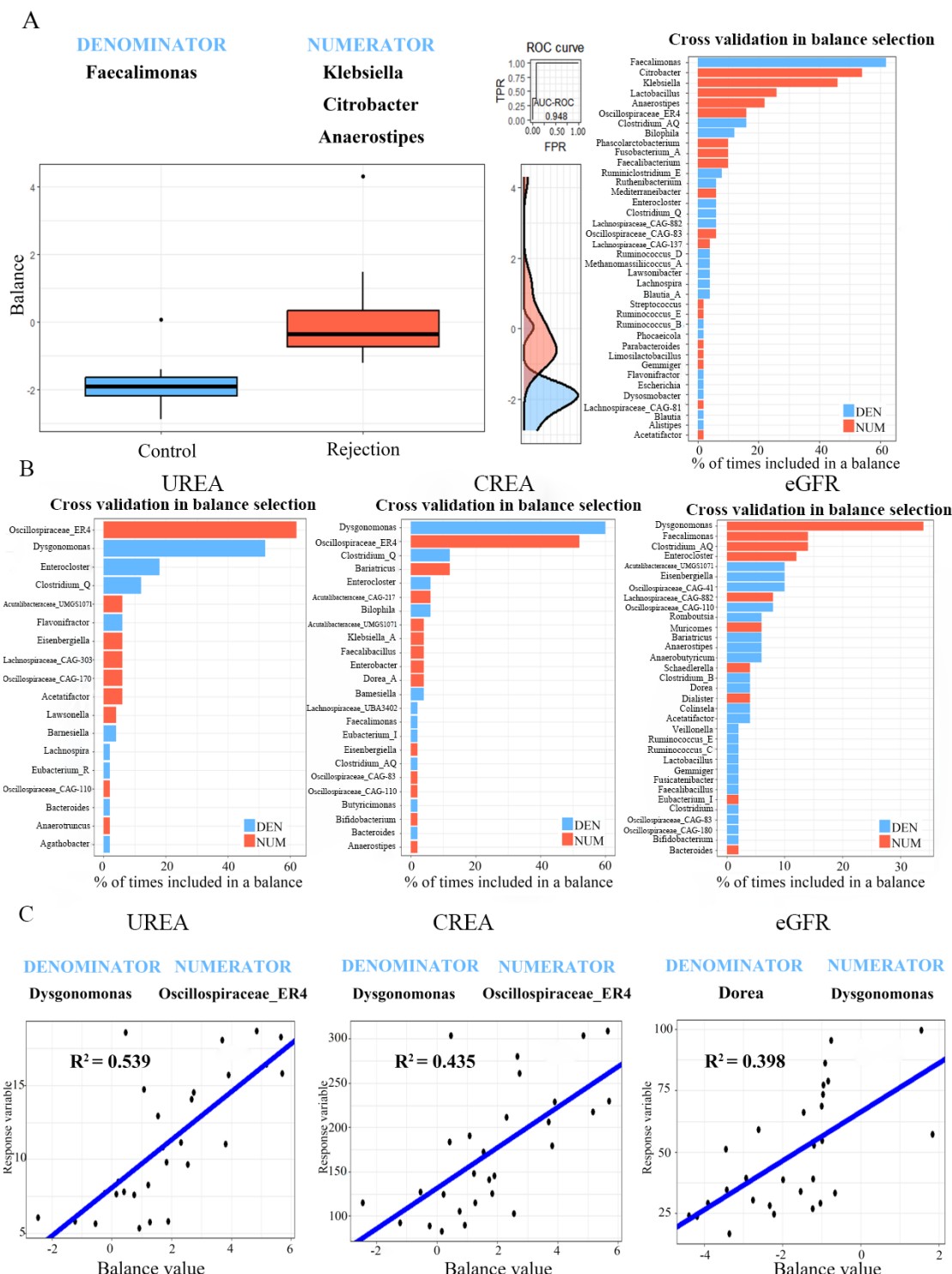

**FIG 4** (A) Description of the global balance for rejection patients. The two groups (numerator and denominator) of taxa that form the global balance are specified at the top of the plot. The intermediate part of the figure contains the ROC curve with its AUC value (0.948) and the density curve for each group. The right part of the figure contains the frequency of gut microbial genus selected in the balance of the cross-validation process in association with rejection patients. (B) The frequency of gut microbial genus selected in the balance of the cross-validation process according to the association with urea, CREA, and eGFR. (C) Association of the balance value with urea, CREA, and eGFR. $R^2$ values were calculated in the regression model.

## Fecal and serum metabolomic profiles at the pathway level

To understand the specific changes in metabolites and annotated pathways, nontargeted metabolomic profiling of serum and fecal samples from the three groups was performed. Metabolites meeting the criteria of FC ≥ 1.2 or FC ≤ 0.83, VIP ≥ 1, and $P < 0.05$ among the three groups were selected (Fig. 5A and B). These metabolites were subsequently annotated to KEGG pathways. The results revealed that among the fecal metabolites, the specific annotated pathways in the rejection group were fatty acid biosynthesis, bile secretion, and histidine metabolism, whereas those in the dysfunctional group were tyrosine metabolism and lysine degradation. Among the serum metabolites, the specific annotated pathways in the rejection group were caffeine metabolism, regulation of lipolysis in adipocytes, and bile secretion, whereas those in the dysfunctional group were biosynthesis of cofactors and folate biosynthesis (Fig. 6A and B). The differentially abundant metabolites between the rejection group and the dysfunctional group compared with the control group were annotated to tryptophan metabolism, which may be a key pathway for distinguishing the control group and these two groups.

## Perturbed gut microbiota interactions with fecal and serum metabolites

An integrated network analysis was performed on the basis of specific differential microbiota and metabolites to investigate the putative mechanisms involved. Between the rejection and control groups, we observed a complex co-occurrence network in which *Agathobaculum butyriciproducens,* 4-pyridoxic acid, niacinamide, tryptamine, and LL-2,6-diaminoheptanedioate were identified as the main contributors to this network. *Agathobaculum butyriciproducens* and *Gemmiger qucibialis* were positively associated with the levels of serum 4-pyridoxic acid, 4-acetamidobutanoate, but negatively associated with the level of fecal tryptamine. These findings indicate that the two species with high abundance in the rejection group were positively linked to metabolites from vitamin digestion and absorption and the arginine and proline metabolism pathways, with the opposite trend for tryptophan metabolism. *Enterobacter cloacae, Enterobacter ludwigii, Klebsiella grimontii,* and *Klebsiella michiganensis* were negatively associated with the levels of fecal D-glutamine and LL-2,6-diaminoheptanedioate from D-amino acid metabolism. Moreover, the genus *Enterobacter,* including *Enterobacter asburiae,* *Enterobacter cloacae* and *Enterobacter kobeiwas,* was positively linked to serum ouabain from bile secretions (Fig. 7A). Between the dysfunction and control groups, *Acetatifactor sp003447295, Anaerostipes hadrus, Bacteroides eggerthii,* and *Eubacterium ventriosum* were positively associated with Homogentisate and L-Dopa from tyrosine metabolism. *Agathobacter rectalisAgathobacter sp900546625*The metabolites Indole-3-ethanol from tryptophan metabolism were negatively associated with *Agathobacter rectalis, Agathobacter sp900546625, Faecalibacterium sp900758465,* and *Faecalibacterium sp900765105* (Fig. 7B).

These findings confirm that gut microbiota dysbiosis, especially enriched levels of *Agathobaculum butyriciproducens* and *Gemmiger qucibialis,* may promote rejection by regulating the characterized metabolites involved in vitamin digestion and absorption, arginine and proline metabolism pathways, and tryptophan metabolism pathways between the gut and the host.

## A noninvasive model for outcome prediction

We aimed to construct a prediction model for the identification of rejection, dysfunction, and control patients and assess the potential value of gut microbial and metabolomic factors as noninvasive diagnostic markers. LASSO regression analysis was used to identify the variables significantly affecting the outcome. Then, the accuracy, calibration curve, and ROC curve of four common clinical models were used to assess the performance, which was based on the specific altered microbiota and metabolites or a combination. The ROC values of the random forest, boost tree, decision tree, and logistic regression

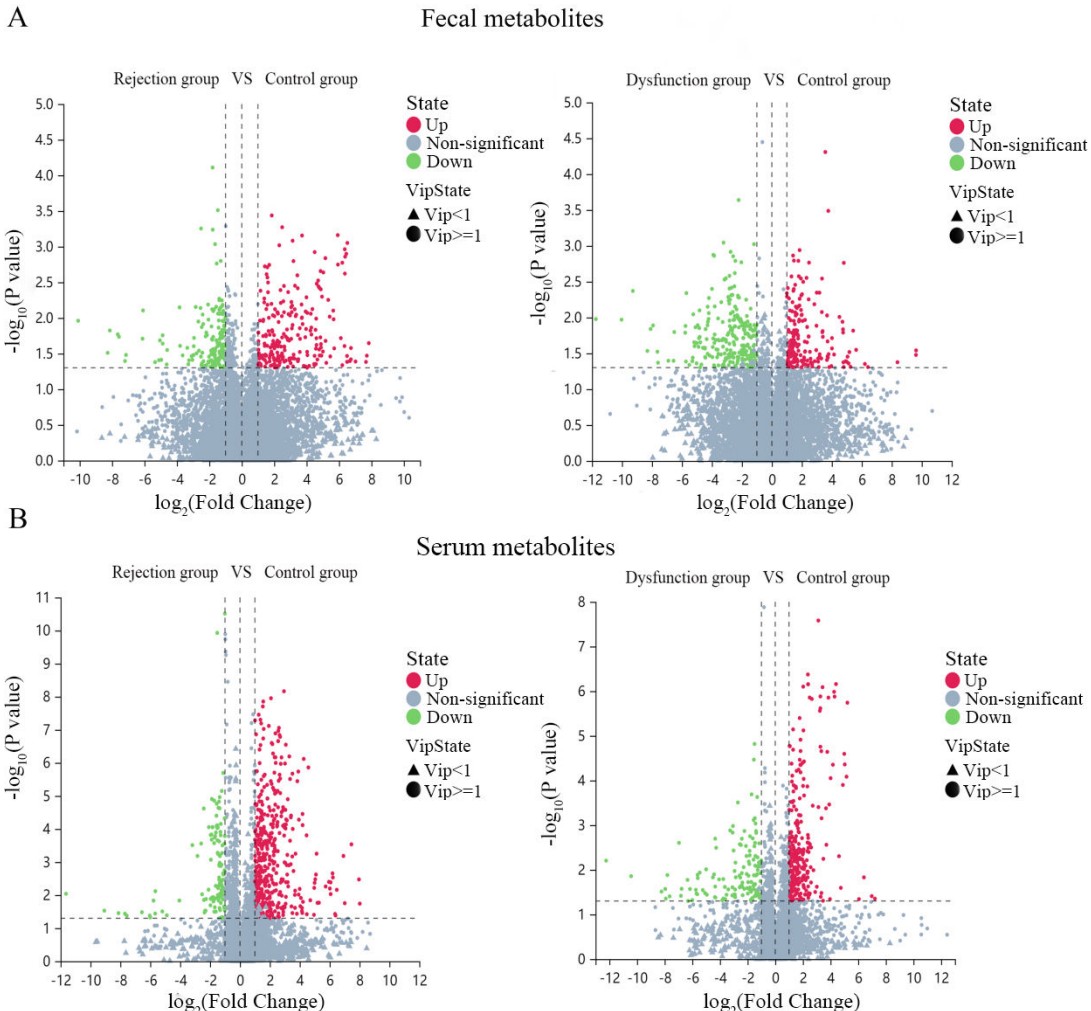

**FIG 5** (A and B) Volcano plot showing the differential fecal and serum metabolites among the three groups. Green is the downregulated significantly differentiated metabolites, red is the upregulated significantly differentiated metabolites, and the non-significant metabolites are gray.

methods for the rejection group based on the combination of microbiota and metabolites were 0.992, 0.926, 0.799, and 0.669 , and for the dysfunction group were 0.942, 0.793, 0.5, 0.886 (Fig. 8), respectively. Obviously, the overall performance of the combination prediction model was better than that of the gut microbiota and metabolites alone for the rejection group, but for the dysfunction group, the ROC values of the four models based on microbiota alone were 0.901, 0.880, 0.5, 0.967, the overall performance was better than that of the combination(Fig. 8).

## DISCUSSION

In summary, the results of integrated multiomics analyses revealed significant differences in the gut microbiome and metabolome features between the rejection and dysfunction groups. The results showed that an imbalance in the gut microbiota can lead to changes in the patient's kidney function and that the altered gut microbiota influences metabolic pathway activity and the abundance of metabolites, with these changes strongly correlated with immunological rejection. This study revealed notable differences in the structure of the gut microbiota; the key constituents of the gut microbiota included *Ruminococcaceae, Faecalibacterium, Agathobacter*, and *Bacteroides*. In the rejection group, *Ruminococcaceae* and *Faecalibacterium* were more enriched, and in the dysfunction group, *Bacteroides* and *Agathobacter* were more enriched. *Ruminococcaceae*

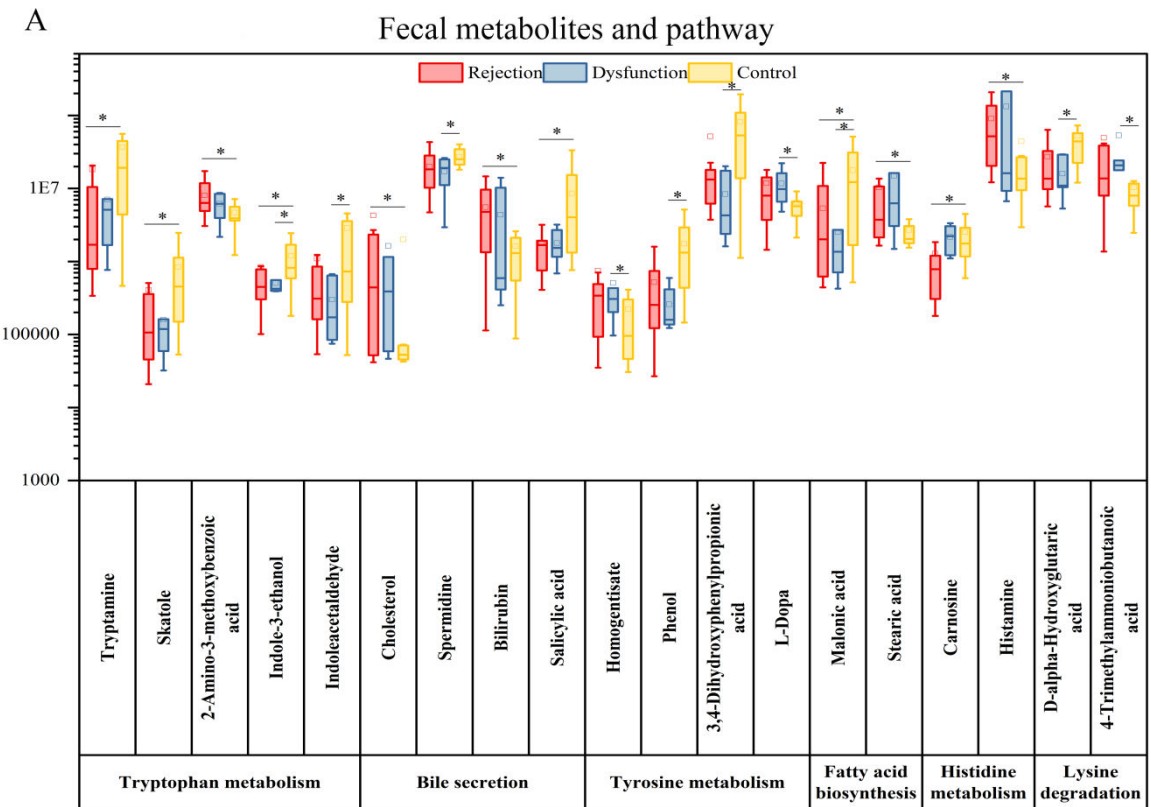

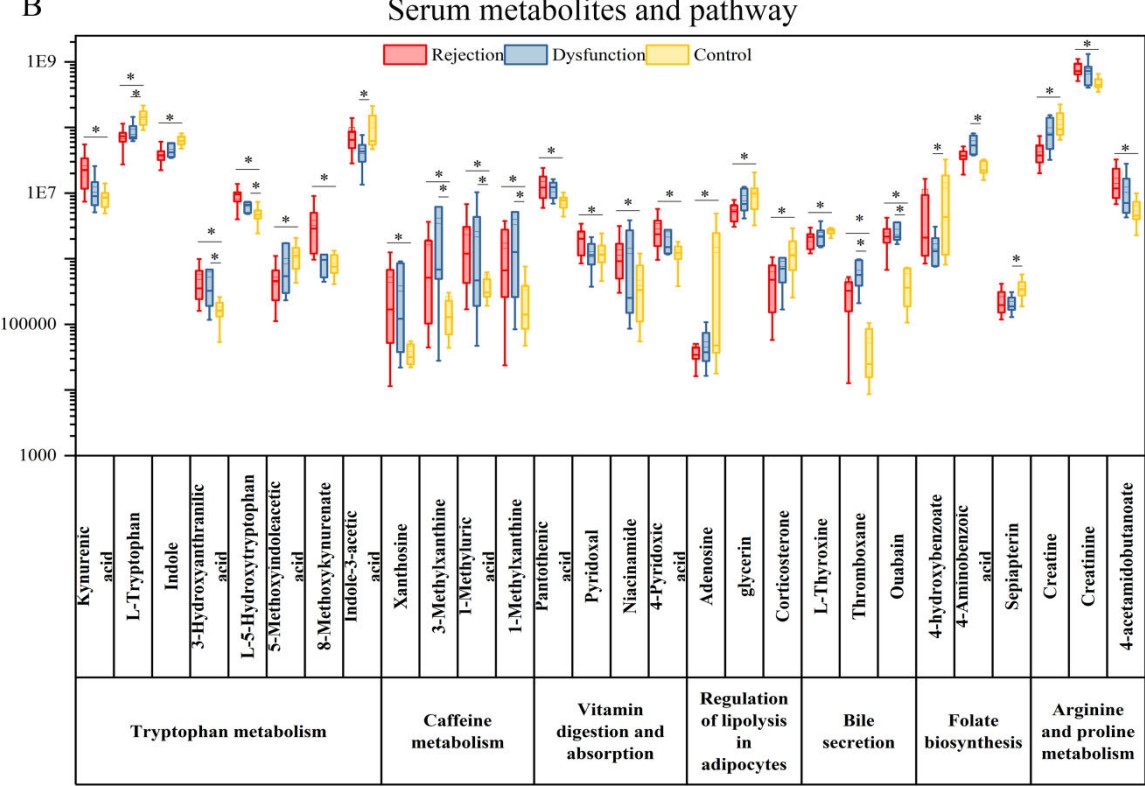

**FIG 6**   (A) The fecal metabolites and their annotated pathway. (B) The serum metabolites and their annotated pathway. *$P < 0.05$.

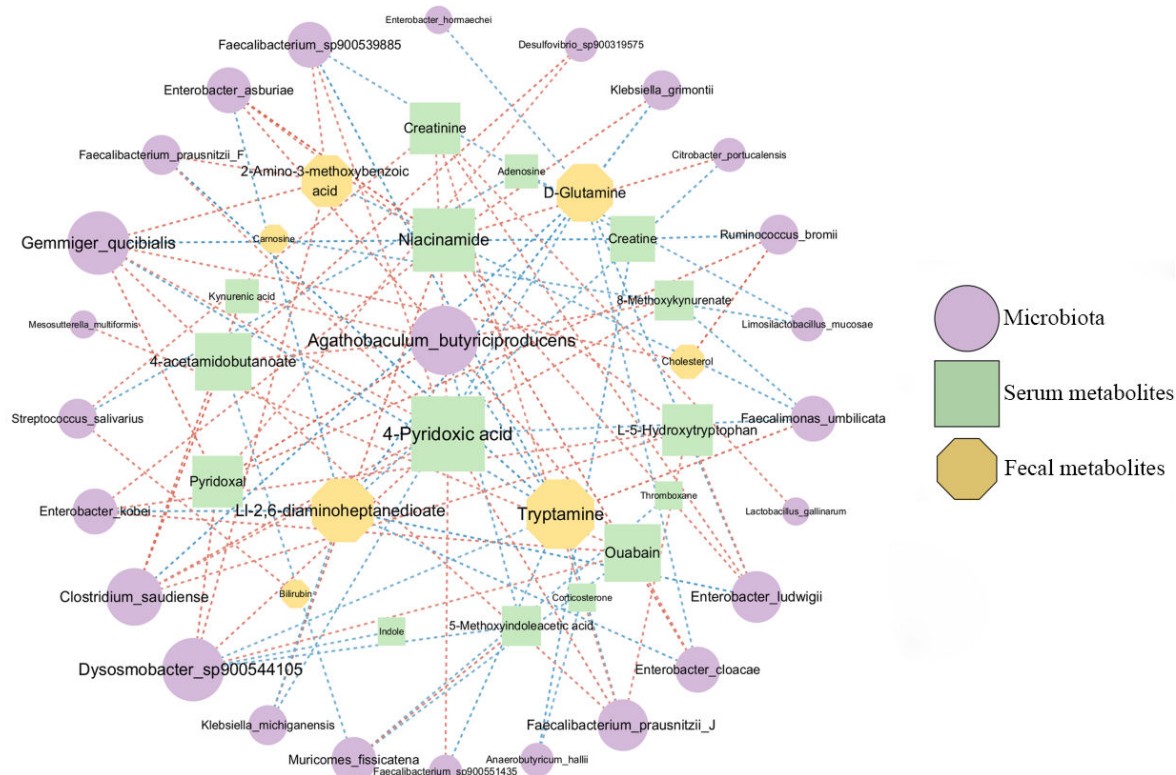

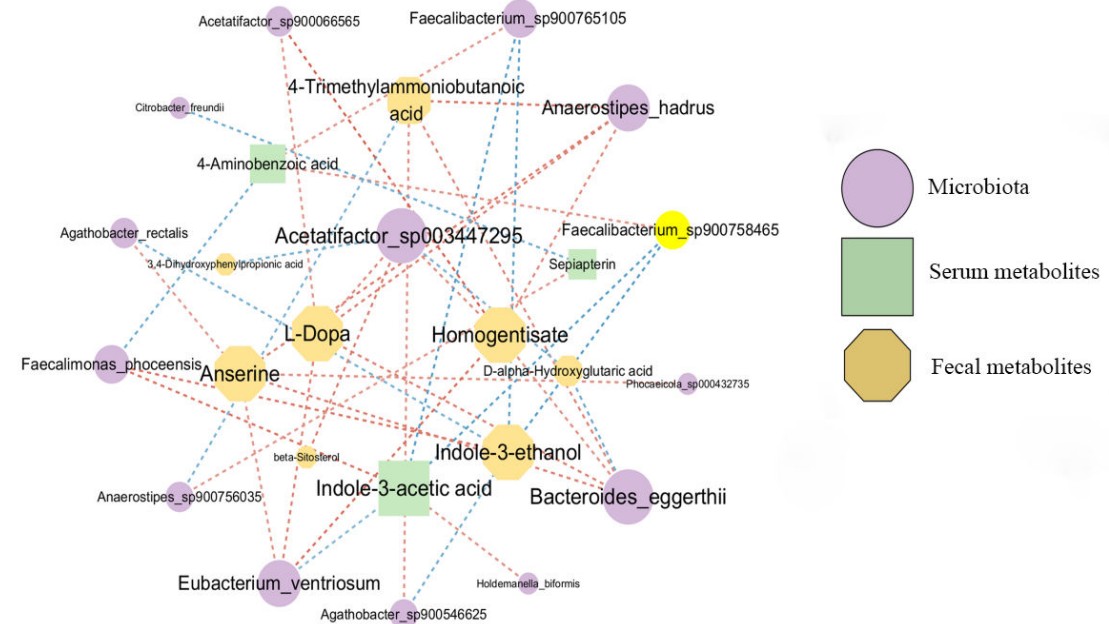

**FIG 7** (A) The co-occurrence network of microbiota and metabolites between the rejection and control groups. (B) The co-occurrence network of microbiota and metabolites between the rejection and control groups. The size of nodes represents the degree of corresponding factors. Direct correlations are indicated as red edges, and inverse correlations as blue edges, with a cutoff of $P < 0.05$ and $r > 0.5$.

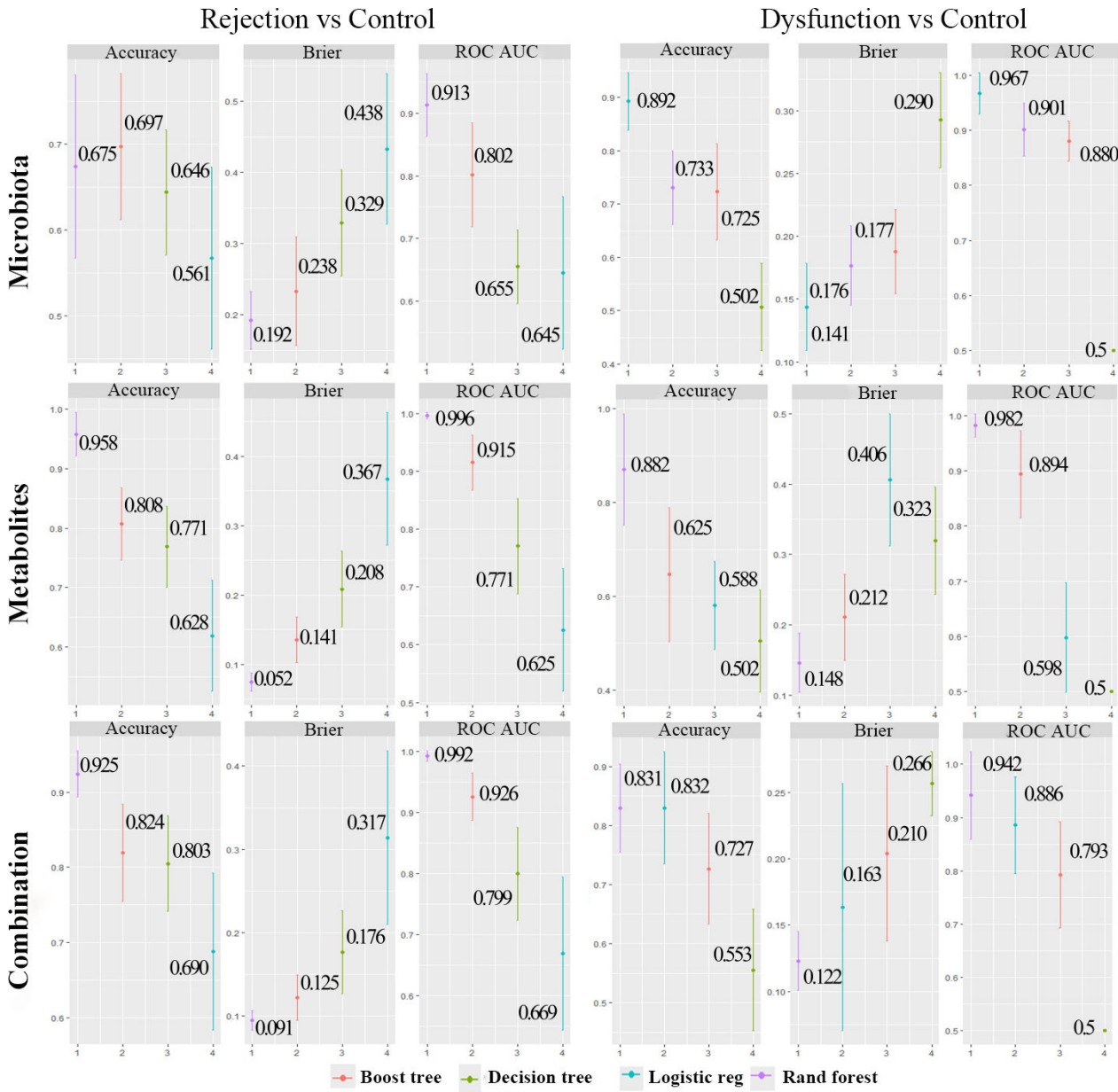

**FIG 8** The diagnostic model of microbiota alone, metabolites alone, and combination for the rejection and dysfunction groups.

and *Faecalibacterium* elicit elevated systemic IgG responses when they translocate across the gut barrier. It may lead to hosts' immune activation (20). These findings suggest that the presence of these microbes in rejection patients might intensify immune responses, causing graft rejection. There was a significant increase in *Bacteroides* and *Agathobacter*, which can produce SCFAs that suppress cellular immunity (17). This suppression might occur through the inhibition of Th1 cell activation, reducing local T-cell infiltration (21). However, there is a contradiction. *Ruminococcaceae* and *Faecalibacterium* are also SCFA-producing species (22, 23). This could be due to the weaker ability of *Ruminococcaceae* and *Faecalibacterium* in SCFA production compared to *Bacteroides*. These contradictions need further mechanistic work to better understand the complex role of such bacterial functions and the resulting host immune responses to these functions. The co-occurrence of gut microbes revealed that dysfunction patients are in an intermediate state of inflammation and immunosuppression, and the SCFA-producing species *Bacteroides*

*ovatus* plays an important role. These findings link changes in the microbiota to immune responses in rejection patients, although specific pathways require further investigation.

An investigation of various metabolite pathways revealed that tryptophan metabolism was changed in both the rejection group and the dysfunction group, suggesting that tryptophan metabolism may be the key pathway of kidney inflammation after transplantation. The correlation between proinflammatory species and tryptamine also supported this conclusion. Tryptamine, one of the primary metabolites annotated in tryptophan metabolism pathways, is converted into various indole metabolites by the *Bacteroides* (24). A decrease in indole metabolites leads to weakened ability to limit Th1 and Th17 differentiation (25). The specific pathways enriched in the rejection group at both the serum and fecal levels were bile secretion and vitamin-related metabolism. Our research shows that ouabain in the bile secretion is enriched in the rejection and dysfunction groups, and ouabain was closely associated with the abundance of proinflammatory species like *Enterobacter cloacae*. Ouabain can regulate the immune system, promote NK cell activity, reduce regulatory T cells, and thus promote the inflammatory response of the graft (26, 27). The co-occurrence network analysis revealed that 4-pyridoxic acid from the vitamin digestion and absorption metabolism, which was a specifically annotated pathway in the rejection group, was the main contributor, and 4-pyridoxic acid was identified as the most sensitive plasma endogenous biomarker of renal organic anion transporters (OAT1/3) (28), and an increase in 4-pyridoxic acid indicates OAT1/3 inhibition, which is associated with a reduced risk of kidney damage (29). The specific pathway in the dysfunction group at the fecal level was tyrosine metabolism. Homogentisate and L-Dopa were the main contributors to the network. L-Dopa was noted to be a biochemical marker of renal dysfunction (30).

Using a diagnostic model, the potential of the gut microbiota and metabolites was demonstrated as a method for diagnosing rejection after renal allograft transplantation. This approach is beneficial for predicting and diagnosing rejection, allowing timely interventions to decrease the probability of rejection. The multiomics approach effectively distinguishes between rejection patients and controls. This distinction can aid outpatient doctors in assessing the status of postoperative patient grafts.

There are some limitations in this study; it is a single-center study. Most of our samples are from Tianjin and the surrounding areas, and each sample collection time is different, so the population variability is unavoidable. The reliability and reproducibility of biomarkers in serum and fecal samples across diverse populations need an external validation. Our metabolomics is non-targeted, and there are enough specific data for some crucial metabolites like SCFA. But this is exactly where we are going next. We may be able to further investigate the mechanism of rejection by targeting the metabolomics of SCFA in the future. Meanwhile, exploring therapeutic interventions targeting specific gut microbiota will be our next target. Eventually, we will collect more samples and test in a larger cohort to enhance the significance of our study.

As a noninvasive approach, multiomics machine learning diagnostic models offer superior compliance with medical standards and enhanced diagnostic performance, as indicated by higher AUC values.

## ACKNOWLEDGMENTS

This study was supported by the Special Funds of the National Natural Science Foundation of China (82241219), the National Major Scientific Research Instrument Development Project of China (82127808), and the Foundation for Innovative Research Groups of the National Natural Science Foundation of China (81921004).

X.D. designed the project, analyzed the data, prepared the figures, and wrote the manuscript. P.-P.Z. and Z.-Y.S. conceived the study, supervised the results, wrote and critically revised the manuscript, and were responsible for its financial support and corresponding work. All the other authors conceived the study and critically revised the manuscript. All the authors approved the final manuscript.

## AUTHOR AFFILIATIONS

[1]First Central Hospital of Tianjin Medical University, Tianjin, China
[2]Department of Kidney Transplantation, Tianjin First Central Hospital, Tianjin, China
[3]Organ Transplantation Research Center, Tianjin First Central Hospital, Tianjin, China
[4]Institute of Transplantation Medicine Nankai University, Nankai University, Tianjin, China
[5]Biological Sample Resource Sharing Center, Tianjin First Central Hospital, Tianjin, China

## AUTHOR ORCIDs

Pan-Pan Zhan  http://orcid.org/0000-0001-6273-6062
Zhong-Yang Shen  http://orcid.org/0009-0000-2212-4539

## FUNDING

| Funder | Grant(s) | Author(s) |
| --- | --- | --- |
| National Natural Science Foundation of China | 82241219 | Zhong-Yang Shen |
| National Major Science and Technology Projects of China | 82127808 | Zhong-Yang Shen |
| National Natural Science Foundation of China | 81921004 | Zhong-Yang Shen |
| Institute of Transplantation Medicine NanKai University | NKTM2023005 | Pan-Pan Zhan |

## AUTHOR CONTRIBUTIONS

Xing Dai, Writing – original draft | Lin Li, Software | Yue-Xin Gao, Data curation | Jian-Xi Wang, Investigation | Yao-Juan Liu, Validation | Ting-Ting Ma, Software | Jian-Ming Zheng, Resources | Pan-Pan Zhan, Writing – review and editing | Zhong-Yang Shen, Writing – review and editing.

## DATA AVAILABILITY

A STORMS (Strengthening The Organizing and Reporting of Microbiome Studies) checklist (2) is available at 10.5281/zenodo.14833288. The data sets have been deposited in NCBI Sequencing Read Archive (accession ID: PRJNA1248590).

## ETHICS APPROVAL

All study participants or their legal guardians provided written informed consent for personal and medical data collection prior to study enrollment. This study was approved by the Science and Technology Ethics Committee of Tianjin First Central Hospital, IRB: 20241230-2.

## ADDITIONAL FILES

The following material is available online.

### Supplemental Material

**Table S1 (mSystems01626-24-s0001.docx).** Baseline characteristics of kidney transplant patients.

### Open Peer Review

**PEER REVIEW HISTORY (review-history.pdf).** An accounting of the reviewer comments and feedback.

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
