## [Reviewer comments · mSystems]

Gut microbiome and metabolome profiles in renal allograft rejection from multiomics integration

Xing Dai, Yu Cao, Lin Li, Yue-Xin Gao, Jian-Xi Wang, Yao-Juan Liu, Ting-Ting Ma, JianMing Zheng, Pan-Pan Zhan, and Zhong-Yang Shen

Corresponding Author(s): Pan-Pan Zhan, Tianjin First Central Hospital

Review Timeline:

Submission Date:	December 4, 2024
Editorial Decision:	January 6, 2025
Revision Received:	February 10, 2025
Accepted:	March 31, 2025

Editor: Li Cui

Reviewer(s): Disclosure of reviewer identity is with reference to reviewer comments included in decision letter(s). The following individuals involved in review of your submission have agreed to reveal their identity: Ali Chaari (Reviewer #1); sohya Aggarwal (Reviewer #2); Aida G. Gabdoulkhakova (Reviewer #3)

Transaction Report:

DOI: <https://doi.org/10.1128/msystems.01626-24>

Re: mSystems01626-24 (**Gut microbiome and metabolome profiles in renal allograft rejection from multiomics integration**)

Dear Ms. Pan-Pan Zhan:

Revision Guidelines

Sincerely,
Li Cui
Editor
mSystems

Reviewer #1 (Comments for the Author):

This paper presents a well-structured and informative analysis of the relationship between gut microbiome, metabolome, and renal allograft rejection. The authors effectively organize their research, beginning with a clear statement of the research problem and objectives, followed by a logical progression through methods, results, and conclusions. The methodology is particularly well-detailed, with a clear description of the patient groupings and the multiomics approach combining metagenomic sequencing and nontargeted metabolomics. The sample size and group distributions are explicitly stated, which adds to the transparency of the research.

However, several key references from recent studies investigating gut microbiota in organ transplantation appear to be missing. Including these would provide important context and strengthen the paper's literature review.

A more detailed discussion of the study's limitations would also strengthen the paper's scientific rigor. A visual representation of the diagnostic models' performance metrics would enhance the presentation of these important findings.

Based on lines 70 and 72 of the paper, I would recommend that the authors be more specific and provide detailed information about:

1. Pro-inflammatory markers:
2. Gut-derived toxic metabolites:

Finally, Looking through the paper excerpt provided, I don't see any mention of an IRB (Institutional Review Board) number. This is a significant oversight as all human studies should clearly state their ethical approval details.

Despite these suggestions for improvement, this study makes a significant contribution to understanding the role of gut microbiota in renal allograft rejection and offers promising clinical applications for diagnosis.

Reviewer #2 (Comments for the Author):

The manuscript titled, "Gut microbiome and 1 metabolome profiles in renal allograft rejection from multiomics integration" provides a detailed and methodologically sound analysis of the gut microbiota and metabolomic profiles associated with renal allograft rejection. It highlights the critical need of transplantation medicine and addresses potential of noninvasive diagnostic approaches, which is an important contribution to the field. Overall the study is well executed but below are some comments and suggestions for improvement:

Authors are suggested to highlight the main metabolites and species in the abstract section.

In the introduction section, authors are suggested to elaborate the major gap in the field and the significance of the study to strengthen the study.

Under methods sections, it would be helpful to the readers to have a concise description of the integrated multiomic framework and why it was chosen. Also, highlight the sample size and how it may affect the generalizability of the findings.

Authors are suggested to give further clarification on how these interactions impact rejection and dysfunction.

In discussion section, while tryptophan metabolism is highlighted, authors are suggested to include other pathways as well if they also holds the significant clinical relevance.

Also, write few statements under discussion section acknowledging the potential limitations, including population variability and the need for external validation.

Also, mentioning some more specific next steps like exploring therapeutic interventions targeting gut microbiota or testing in larger cohort would also enhance the significance of the study.

Reviewer #3 (Comments for the Author):

The issue related to the problem of kidney transplant rejection and possible interaction with gut microbiota and bacterial metabolites. The authors revealed the dysbiosis in the gut of the patients with rejection and dysfunction of the kidney transplant; and they suggested that it can lead to changes in the patient's kidney function and that the altered gut microbiota influences metabolic pathway activity and the abundance of metabolites. Also, the authors suggested that these changes strongly correlated with immunological activation and rejection. Data analysis demonstrated that bacteria Ruminococcaceae, Faecalibacterium, and Agathobacter are enriched in the gut of the patients with kidney rejection and dysfunction. The analysis has shown high correlation with amino acid and bile acid metabolism, possible immunomodulatory role of these metabolites on kidney function.

1) In the aspect of metabolites, SCFA has an important role to play, especially since changes in the number of butyrate-producing bacteria were detected, so it would be appropriate to include the estimation of acetate, butyrate and propionate levels in the samples.

2) The bacteria Ruminococcaceae, Faecalibacterium and Agathobacter enriched in the groups of rejection and graft dysfunction are generally considered as useful butyrate-producing bacteria. Therefore, the discussion in this part seems somewhat contradictory - the association of these bacteria with pro-inflammatory properties with activation of CD8/CD4 positive T lymphocytes and enhanced immunosuppressive activity (lines 279-282, 332-333). How did change the proportion of total SFCA-producing bacteria in the groups?

3) There are data from other researchers on the gut microbiome during renal transplantation, what is common and what is specific for your population?

4) What is the role of the donor kidney microbiota in transplantation?

Minor remarks:

- 1) Line 191 - Probably was meant "phylum" instead of family
- 2) Line 218
- 3) Fig.4(B) - typing error in the figure legend - must be "genus"
- 4) Fig. 4 (B) - very small fonts
- 5) Fig.5 - must be "green".

The manuscript titled, “**Gut microbiome and 1 metabolome profiles in renal allograft rejection from multiomics integration**” provides a detailed and methodologically sound analysis of the gut microbiota and metabolomic profiles associated with renal allograft rejection. It highlights the critical need of transplantation medicine and addresses potential of noninvasive diagnostic approaches, which is an important contribution to the field. Overall the study is well executed but below are some comments and suggestions for improvement:

Authors are suggested to highlight the main metabolites and species in the abstract section.

In the introduction section, authors are suggested to elaborate the major gap in the field and the significance of the study to strengthen the study.

Under methods sections, it would be helpful to the readers to have a concise description of the integrated multiomic framework and why it was chosen. Also, highlight the sample size and how it may affect the generalizability of the findings.

Authors are suggested to give further clarification on how these interactions impact rejection and dysfunction.

In discussion section, while tryptophan metabolism is highlighted, authors are suggested to include other pathways as well if they also hold the significant clinical relevance.

Also, write few statements under discussion section acknowledging the potential limitations, including population variability and the need for external validation.

Also, mentioning some more specific next steps like exploring therapeutic interventions targeting gut microbiota or testing in larger cohort would also enhance the significance of the study.

Response to reviewer 1:

We sincerely thank the editor and you for the valuable feedback, we consider the advices carefully and have used to improve the quality of our manuscript. Your comments are laid out below in italicized font and specific concerns have been numbered. Our response is given in normal font and changes/additions to the manuscript are given in the blue text.

1. several key references from recent studies investigating gut microbiota in organ transplantation appear to be missing.

We have added background information, Now, There are several types of non-invasive biomarker technologies (metagenomics, metabolomic, epigenetic, proteomic, and transcriptomic) currently used, but it is not clear which method is more accurate. Recent study revealed that gut microbiota is associated with organ post-transplant complications, like diarrhea, urinary tract infection, rejection and mortality [3-7]. Obviously, Gut microbiota plays an important role after solid organ transplantation, it may become an insightful biomarker to predict graft rejection, because gut microbiota can modulate immune system by metabolites to affect graft function [8,9]. **(lines 71 to 78 of the paper)**

2. A more detailed discussion of the study's limitations would also strengthen the paper's scientific rigor

Based on your suggestions, we have added a more detailed discussion of the study's limitations. There are some limitations in this study, it is a single-center study, most of our samples are from Tianjin and the surrounding areas, and each sample collection time is different, so the population variability is unavoidable, and The reliability and reproducibility of biomarkers in serum and fecal samples across diverse populations need for a external validation. Our metabolomics is non-targeted and there are enough specific datas for some crucial metabolites like SCFA.

But this is exactly where we are going next, and we may be able to further investigate the mechanism of rejection by targeting the metabolomics of SCFA in the future, and meanwhile exploring therapeutic interventions targeting specific gut microbiota will be our next target. Eventually we collect more samples and test in larger cohort to enhance the significance of our study. **(lines 405 to 415 of the paper)**

3. *A visual representation of the diagnostic models' performance metrics would enhance the presentation of these important findings.*

We marked the ROC curve values on our figure (Figure 1).

Figure 1

4. Based on lines 70 and 72 of the paper, I would recommend that the authors be more specific and provide detailed information about:

1. Pro-inflammatory markers: IL-1, IL-6 and IL-18

2. Gut-derived toxic metabolites: Phenols and indoles

This dysregulation induces a proinflammatory response in intestinal epithelial cells. The IL-1, IL-6 and IL-18 secreted by intestinal epithelial cells triggered to eliminate pathogens. Dendritic cells, and macrophages induce development of the effector CD4⁺ T cells TH1 and TH17 by secreting IL-10 and IL-23^[12]. Simultaneously, gut-derived toxic metabolites---phenols and indoles are further metabolized into P-cresol sulfate (PCS) (lines 83 to 88 of the paper)

5. IRB (Institutional Review Board) number

The research protocol and procedures were approved by the Tianjin First Central Hospital Ethical Committee (Approval No. 20241230-2) (lines 29 to 30 of the paper)

Response to reviewer 2:

We feel great thanks for your professional review work on our article. As you are concerned, there are several problems that need to be addressed. According to your nice suggestions, we have made extensive corrections to our previous draft, the detailed corrections are listed below. Your comments are laid out below in italicized font and specific concerns have been numbered. Our response is given in normal font and changes/additions to the manuscript are given in the blue text.

1. Authors are suggested to highlight the main metabolites and species in the abstract section

Based on your suggestions, we have highlighted the main metabolites and species in the abstract section as follows and made modification on the revised manuscript.

Results: The microbiota composition and the related genes in the rejection group significantly differed from that in the dysfunction and control groups at the phylum, genus and species levels ($P < 0.001$). *The core species in the rejection group networks were Escherichia_coli and Ruminococcus_gnavus, while core species in the dysfunction group networks were Faecalibacterium_prausnitzii, Bacteroides_ovatus.*

The balance of specific microbial species was associated with kidney function in rejection patients. Spearman analysis revealed that specific differential species like *Agathobaculum_butyriciproducens*, *Gemmiger_qucibialis* were closely linked to the levels of serum 4-pyridoxic acid, 4-acetamidobutanoate, creatine and fecal tryptamine from specific differential pathways. Finally, we constructed four clinical models to distinguish the rejection and dysfunction groups, and the model had excellent diagnostic performance. **(lines 44 to 53 of the paper)**

2. In the introduction section, authors are suggested to elaborate the major gap in

the field and the significance of the study to strengthen the study

We have added the following text to the background section.

Now, There are several types of non-invasive biomarker technologies (metagenomics, metabolomic, epigenetic, proteomic, and transcriptomic) currently used, but it is not clear which method is more accurate. Recent study revealed that gut microbiota is associated with organ post-transplant complications, like diarrhea, urinary tract infection, rejection and mortality^[3-7] . Obviously, Gut microbiota plays an important role after solid organ transplantation, it may become an insightful biomarker to predict graft rejection, because gut microbiota can modulate immune system by metabolites to affect graft function^[8,9]**(lines 71 to 78 of the paper)**

Because of the strong correlation between the gut microbiota and metabolites, the combination of microbiota and metabolomic may improve the accurate of diagnosis of allograft rejection. **(lines 91 to 93 of the paper)**

3.Undre methods sections, it would be helpful to the readers to have a concise description of the integrated multiomic framework and why it was chosen.

Based on your suggestions, we added the part of multiomics integration to the method section as follows.

Multiomics integration

The network was drawn by spearman correlation analysis to reveal the correlation between gut microbiota and metabolites. Gut microbiota and metabolites annotated pathway were linked by the network. On the one hand, It helps to explore the mechanisms by which the gut microbiota affects rejection reaction by metabolites, and the predictive effect of combined markers on rejection was analyzed.**(lines 185 to 190 of the paper)**

4. Also, highlight the sample size and how it may affect the generalizability of the findings

We have been calculated the sample size by PASS (version 2020). We set α to 0.05 and power to 0.09, the mean relative abundance of *Clostridia*, *Bacilli*, *Clostridiales*, *Lactobacillales*, *Faecalibacterium* in rejection group were 0.616, 0.026, 0.616, 0.020, 0.136 respectively, while in control groups were 0.414, 0.082, 0.414, 0.081, 0.071, estimated mean σ were 0.15, 0.04, 0.15, 0.04, 0.05. The calculated total sample size were 26, 24, 26, 22, 28 respectively. We have met the demand of sample size, indicated that our findings are reliable. However, a larger sample size and multicenter research will be more conducive to generalization of the findings. The data used are cited from Wang's research (Wang J, et al., Shifts in Intestinal Metabolic Profile Among Kidney Transplantation Recipients with Antibody-Mediated Rejection. *Ther Clin Risk Manag.* 2023 Mar 3;19:207-217).

5. Authors are suggested to give further clarification on how these interactions impact rejection and dysfunction

The question you mentioned is a hot research topic at present. The most common opinion is that SCFA-producing bacteria inhibits graft rejection through immune regulation. In our study, the proportion of SCFA-producing bacteria decreased in the rejection group, this may be one of the mechanisms of rejection. In addition, our own research found that the tryptophan metabolism pathway may play a important role in rejection.

Tryptamine, one of the primary metabolites annotated in tryptophan metabolism pathways, it is converted into various Indole metabolites by the *Bacteroides*^[25], the decrease of indole metabolites lead to weakened ability to limit Th1 and Th17 differentiation^[26]. The specific pathways enriched in the rejection group at both the serum and fecal levels were Bile secretion, lipid metabolism and Vitamin-related metabolism, Our research shows that ouabain in the Bile secretion enriched in the

rejection and dysfunction group, and ouabain was closely associated with the abundance of proinflammatory species like *Enterobacter_cloacae*. The ouabain can regulate immunity system, promote NK cell activity, reduced regulatory T cells, and thus promote the inflammatory response of the graft^[27,28]. **(we added this in the discussion section, lines 379 to 388 of the paper)**

6.authors are suggested to include other pathways as well if they also holds the significant clinical relevance

We added ouabain in the Bile secretion metabolism

Our research shows that ouabain in the bile secretion enriched in the rejection and dysfunction group, and ouabain was closely associated with the abundance of proinflammatory species like *Enterobacter_cloacae*. The ouabain can regulate immunity system, promote NK cell activity, reduced regulatory T cells, and thus promote the inflammatory response of the graft^[27,28]. **(lines 384 to 388 of the paper).**

We have added the clinical relevance of the Vitamin-related metabolism and Tyrosine metabolism in the discussion section.

The co-occurrence network analysis revealed that 4-pyridoxic acid from the Vitamin digestion and absorption metabolism was the main contributor, and 4-pyridoxic acid (PDA) was identified as the most sensitive plasma endogenous biomarker of renal organic anion transporters (OAT1/3)^[29], An increase in 4-pyridoxic acid indicates OAT1/3 inhibition, resulting in a reduced ability to suffer kidney damage. The specific pathway varied in the rejection group^[30]. The specific pathway in the dysfunction group at the fecal level was Tyrosine metabolism. Homogentisate and L-DOPA were the main contributors to the network. L-DOPA was noted to be a biochemical marker of renal dysfunction^[31]. **(lines 388 to 396 of the paper).**

7. Also, write few statements under discussion section acknowledging the potential limitations, including population variability and the need for external validation.

There are some limitations in this study, it is a single-center study, most of our samples are from Tianjin and the surrounding areas, and each sample collection time is different, so the population variability is unavoidable, and The reliability and reproducibility of biomarkers in serum and fecal samples across diverse populations need for a external validation (lines 405 to 409 of the paper).

8. Also, mentioning some more specific next steps like exploring therapeutic interventions targeting gut microbiota or testing in larger cohort would also enhance the significance of the study.

We may be able to further investigate the mechanism of rejection by targeting metabolomics in the future, targeting metabolomics may help us further understand the relationship among gut microbiota, metabolites (like SCFA) and rejection. Meanwhile therapeutic interventions targeting specific gut microbiota will be our next target. Eventually we collect more samples and test them in larger cohort to enhance the significance of our study.

Response to reviewer 3:

On behalf of all the contributing authors, we would like to express our sincere appreciations of your letter and constructive comments concerning our article. These comments are all valuable and helpful for improving our article. Your comments are laid out below in italicized font and specific concerns have been numbered. Our response is given in normal font and changes/additions to the manuscript are given in the blue text.

1. In the aspect of metabolites, SCFA has an important role to play, especially since changes in the number of butyrate-producing bacteria were detected, so it would be appropriate to include the estimation of acetate, butyrate and propionate levels in the samples.

Your question is a hot topic right now, SCFA has been shown to inhibits rejection by modulating immunity, focusing on the number of SCFA can helps us further understand the mechanism of kidney transplant rejection. Unfortunately, our metabolomics is non-targeted and there are no specific data for acetate, butyrate and propionate. **We can only list the SCFA-producing genus (This also answers your question: How did change the proportion of total SFCA-producing bacteria in the groups)**

The top 30 genus in relative abundance among three groups are listed below.

category	Rejection	Dysfunction	Control
Phocaeicola	14.0097	24.9278	17.7158
Bacteroides (Produce SCFA)	10.3908	26.4434	20.7834
Escherichia	9.0041	3.637	3.7173
Lachnospira(Produce SCFA)	2.9779	4.2708	8.2972
Faecalibacterium(Produce SCFA)	5.7829	4.491	2.3636
Parabacteroides(Produce SCFA)	2.8269	0.992	6.5124
Methanomassiliicoccus_A	3.2285	3.4585	2.6056
Blautia_A(Produce SCFA)	2.5743	1.0844	2.9081
Prevotella	4.6204	0.3893	0.5691
Enterocloster	1.5379	1.7326	3.1588
Roseburia(Produce SCFA)	2.1151	2.0124	2.1661
Alistipes(Produce SCFA)	1.2072	1.0742	2.6259
Acetatifactor(Produce SCFA)	1.7649	2.4267	0.7329
Agathobacter(Produce SCFA)	0.6133	3.8796	0.5094
Gemmiger(Produce SCFA)	1.8798	0.2756	0.7433
Citrobacter	2.2689	0.2548	0.0522
Megamonas(Produce SCFA)	1.3409	0.1032	1.1129

Dysosmobacter	0.9624	0.3538	1.3577
Klebsiella	1.8506	0.4061	0.1024
Lactobacillus(Produce SCFA)	1.9467	0.0016	0.002
Mediterraneibacter	1.5636	0.2565	0.3599
Phascolarctobacterium(Produce SCFA)	1.14	0.6758	0.3336
Flavonifractor	0.6338	0.4081	1.0199
Veillonella(Produce SCFA)	0.8024	0.5895	0.4904
Ruminococcus_B(Produce SCFA)	0.4419	0.6406	0.7598
CAG-882	0.0163	0.0654	1.5386
Fusobacterium_A	0.4995	0.222	0.7375
Fusicatenibacter	0.5294	0.3181	0.6015
Streptococcus	0.6636	0.4256	0.1879
CAG-495	0.0038	2.1563	0.0138
Other	20.8026	12.0273	15.9211

A total of 15 SCFA-producing genus in the table, it reveals that the proportion of SCFA-producing in the three groups are 37.8%(Rejection group), 49.0%(Dysfunction group) and 50.3%(Control group), The largest proportion change in the rejection group is *Bacteroides*, it may mean that *Bacteroides* plays a large role in anti-inflammatory.

2. The bacteria Ruminococcaceae, Faecalibacterium and Agathobacter enriched in the groups of rejection and graft dysfunction are generally considered as useful butyrate-producing bacteria. Therefore, the discussion in this part seems somewhat contradictory-the association of these bacteria with pro-inflammatory properties with activation of CD8/CD4 positive T lymphocytes and enhanced immunosuppressive activity (lines 279-282, 332-333).

We did not discuss this part clearly, and we have made corrections.

In the Rejection group, *Ruminococcaceae, Faecalibacterium* were more enriched and in the dysfunction group, *Bacteroides* and *Agathobacter* were more enriched. *Ruminococcaceae, Faecalibacterium* elicit elevated systemic IgG responses when they translocate across the gut barrier, it may lead to hosts immune activation^[21]. These findings suggest that the presence of these microbes in rejection patients might intensify immune responses, causing graft rejection. A significant increase in *Bacteroides, Agathobacter*, which can produce short-chain fatty acids (SCFAs) that suppress cellular

immunity^[18], This suppression might occur through the inhibition of Th1 cell activation, reducing local T-cell infiltration^[22]. However, there is a contradiction, *Ruminococcaceae* and *Faecalibacterium* are also SCFA-producing species^[23,24]. This could be the ability of *Ruminococcaceae*, *Faecalibacterium* in SCFA-producing is weaker than *Bacteroides*. These contradictions need further mechanistic work to better understand the complex role of such bacterial functions, and resulting host immune responses to these functions. **(We add this part in our discussion, lines 356 to 369 of the paper)**.

3. There are data from other researchers on the gut microbiome during renal transplantation, what is common and what is specific for your population?

We compared several previous studies, The common gut microbiota include *Firmicutes*, *Bacteroides*, *Proteobacteria*, *Clostridia*, *Enterobacteriaceae*, *Ruminococcus*, *Faecalibacterium*, *Escherichia*, *Streptococcus*, *Lactobacillus*^[1-8]. **(References are at the end of this text file)**

Our specific gut microbiota: *Phocaeicola*, *Gemmiger_qucibialis*, *Agathobacter*, *Erysipelotrichales*, *Dysgonomonadaceae*, and *Faecalimons*. The *Phocaeicola* (recently reclassified from *Bacteroides*) and *Agathobacter* were SCFA-producing species^[9, 10], **(References are at the end of this text file)** *Gemmiger_qucibialis* increases in the rejection group, were closely linked to metabolites from Vitamin digestion and absorption and the Arginine and proline metabolism pathways, with the opposite trend for tryptophan metabolism. It may play a important role in the rejection. *Erysipelotrichales*, *Dysgonomonadaceae*, and *Faecalimons* were more enriched in the control group, the function of three species is not clear yet and need to be explored.

4. What is the role of the donor kidney microbiota in transplantation?

There was no significant alterations in gut microbiota richness, diversity, composition and function of the donor kidney after nephrectomy^[11]. The study revealed Pre-transplantation microbial similarity in unrelated donors and recipients may be

associated with 6-month allograft function^[12].

If the donor kidney carries microbiota, it should increase the chance of postoperative infection of the recipient, but there is no relevant literature.

The detection time of gut microbiota in our samples was over 2 years after kidney. It is still unclear whether the gut microbiota of the donor has any influence on our samples, but this time may weaken the impact on the gut microbiota before transplantation to a relatively low level.

5. Minor remarks:

- 1) Line 191 - Probably was meant "phylum" instead of family;
- 2) Line 218;
- 3) Fig.4(B) - typing error in the figure legend - must be "genus";
- 4) Fig. 4 (B) - very small fonts;
- 5) Fig.5 - must be "green".

All are revised as required

Reference

1. YE Guirong*, Z.M., YU Lixin, YE Junsheng, YAO Lin, SHI Lisha, <*Gut microbiota in renal transplant recipients, patients with chronic kidney disease and healthy subjects.pdf*>. J South Med Univ, 2018. **38(12): 1401-1408.**
2. Stepkowski, S., et al., *Impact of maintenance immunosuppressive therapy on the fecal microbiome of renal transplant recipients: Comparison between an everolimus- and a standard tacrolimus-based regimen.* Plos One, 2017. **12(5).**
3. Lee, J.R., et al., *Butyrate-producing gut bacteria and viral infections in kidney transplant recipients: A pilot study.* Transplant Infectious Disease, 2019. **21(6).**
4. Lee, J.R., et al., *Gut microbiota dysbiosis and diarrhea in kidney transplant recipients.* American Journal of Transplantation, 2019. **19(2): p. 488-500.**
5. Swarte, J.C., et al., *Characteristics and Dysbiosis of the Gut Microbiome in Renal Transplant Recipients.* Journal of Clinical Medicine, 2020. **9(2).**
6. Yu, D.H., et al., *The Alteration human of gut microbiota and metabolites before and after renal transplantation.* Microbial Pathogenesis, 2021. **160.**
7. Fricke, W.F., et al., *Human Microbiota Characterization in the Course of Renal Transplantation.* American Journal of Transplantation, 2014. **14(2): p. 416-427.**
8. Samuel Chan , M.M.C.M.H., Scott B Campbell , Ross S Francis , Nicole M Isbel, Elaine M Pascoe, David W Johnson, *Characteristics of the gastrointestinal microbiota in paired live kidney donors and recipients.* Nephrology (Carlton), 2021. **26(5):471-478.**
9. García-López, M., et al., *Analysis of 1,000 Type-Strain Genomes Improves Taxonomic Classification of Bacteroidetes.* Frontiers in Microbiology, 2019. **10.**
10. Jing, Y., et al., *Spinal cord injury-induced gut dysbiosis influences neurological recovery partly through short-chain fatty acids.* npj Biofilms and Microbiomes, 2023. **9(1).**
11. Chan, S., et al., *Characteristics of the gastrointestinal microbiota in paired live kidney donors and recipients.* Nephrology (Carlton), 2021. **26(5): p. 471-478.**
12. Kim, J.E., et al., *Effect of the similarity of gut microbiota composition between donor and recipient on graft function after living donor kidney transplantation.* Scientific Reports, 2020. **10(1).**

Re: mSystems01626-24R1 (**Gut microbiome and metabolome profiles in renal allograft rejection from multiomics integration**)

Dear Ms. Pan-Pan Zhan:

Your manuscript has been accepted, and I am forwarding it to the ASM production staff for publication. Your paper will first be checked to make sure all elements meet the technical requirements. ASM staff will contact you if anything needs to be revised before copyediting and production can begin. Otherwise, you will be notified when your proofs are ready to be viewed.

Sincerely,
Li Cui
Editor
mSystems

Reviewer #1 (Comments for the Author):

The authors have addressed all the recommendations, and as a result, the paper has significantly improved

Reviewer #3 (Comments for the Author):

It was my pleasure to review the manuscript. The amendments improved the text and made the data clear to understand. The table of the top 30 SFCA-producing genus in relative abundance among the studied groups, that authors compiled in the response to my comment, could be included as Supplementary material, but this is at the authors' discretion.

This paper, titled "Gut Microbiome and Metabolome Profiles in Renal Allograft Rejection from Multiomics Integration," offers a comprehensive analysis of the interplay between gut microbiota and metabolomic profiles in a progression of renal allograft rejection. The authors presented a well-structured investigation with a clear articulation of the research problem and objectives. A multiomics approach including metagenomic sequencing, nontargeted metabolomics and appropriated bioinformative analysis allowed revealing taxons which specific for rejection and dysfunction groups. This study underscores the importance of transplantation medicine and explores the potential for noninvasive diagnostic methods.

The amendments improved the text of the article and made the data clear to understand. The graphical materials are clear. The table of the top 30 SFCA-producing genus in relative abundance among the studied groups, that authors compiled in the response to my comment, could be included as Supplementary material, but this is at the authors' discretion. In this form, the article can be accepted for publication.

Aida Gabdoulkhakova, PhD, Research Associate,
Laboratory of Gene and Cellular Technologies,
Institute of fundamental medicine and biology
Kazan Federal University
AiGGabdulhakova@kpfu.ru